# Midlife socioeconomic position and old-age dementia mortality: a large prospective register-based study from Finland

Kaarina Korhonen [ID],[1] Elina Einiö,[1,2,3] Taina Leinonen,[4] Lasse Tarkiainen,[1] Pekka Martikainen[1,3,5]

[1]Population Research Unit, Faculty of Social Sciences, University of Helsinki, Helsinki, Finland
[2]Department of Social Policy, London School of Economics and Political Science, London, United Kingdom
[3]Max Planck Institute for Demographic Research, Rostock, Germany
[4]Finnish Institute of Occupational Health, Helsinki, Finland
[5]Department of Public Health Sciences, Stockholm University, Stockholm, Sweden

**Correspondence to**
MSocSci Kaarina Korhonen;
kaarina.korhonen@helsinki.fi

## ABSTRACT

**Objectives** To assess the association between multiple indicators of socioeconomic position and dementia-related death, and to estimate the contribution of dementia to socioeconomic differences in overall mortality at older ages.

**Design** Prospective population-based register study.

**Setting** Finland.

**Participants** 11% random sample of the population aged 70–87 years resident in Finland at the end of year 2000 (n=54 964).

**Main outcome measure** Incidence rates, Kaplan-Meier survival probabilities and Cox regression HRs of dementia mortality in 2001–2016 by midlife education, occupational social class and household income measured at ages 53–57 years.

**Results** During the 528 387 person-years at risk, 11 395 individuals died from dementia (215.7 per 10 000 person-years). Lower midlife education, occupational social class and household income were associated with higher dementia mortality, and the differences persisted to the oldest old ages. Compared with mortality from all other causes, however, the socioeconomic differences emerged later. Dementia accounted for 28% of the difference between low and high education groups in overall mortality at age 70+ years, and for 21% of the difference between lowest and highest household income quintiles. All indicators of socioeconomic position were independently associated with dementia mortality, low household income being the strongest independent predictor (HR=1.24, 95% CI 1.16 to 1.32), followed by basic education (HR=1.14, 1.06 to 1.23). Manual occupational social class was related to a 6% higher hazard (HR=1.06, 1.01 to 1.11) compared with non-manual social class. Adjustment for midlife economic activity, baseline marital status and chronic health conditions attenuated the excess hazard of low midlife household income, although significant effects remained.

**Conclusion** Several indicators of socioeconomic position predict dementia mortality independently and socioeconomic inequalities persist into the oldest old ages. The results demonstrate that dementia is among the most important contributors to socioeconomic inequalities in overall mortality at older ages.

### Strengths and limitations of this study

► We used longitudinal registry data that permit a 15-year follow-up of dementia mortality with no attrition or recall bias.

► Dementia is documented in the National Death Register with high specificity.

► Due to the use of register data, traditional dementia risk factors such as smoking and physical activity could not be measured.

► All indicators of socioeconomic position were measured in midlife in order to avoid selection to socioeconomic groups on the basis of cognitive decline.

► This is the first study to show the contribution of dementia to the socioeconomic inequalities in overall mortality at older ages.

## INTRODUCTION

Socioeconomic inequality in health and mortality is one of the most consistent findings in the demographic and social epidemiological literature. Lower education, occupational social class and income are strong predictors of all-cause and cause-specific mortality particularly among the working-age population, but inequalities are clear also at older ages.[1–4] Among the ageing population, the key factors affecting morbidity and disability are Alzheimer's disease and other forms of progressive dementia. Globally, an estimated 47 million people lived with dementia in 2015, and the number is projected to triple by 2050.[5] In England and Wales, dementia has already become the leading cause of death.[6] Despite the growing societal impact, however, no comprehensive understanding exists about the socioeconomic patterns of dementia mortality.

Educational inequalities in dementia mortality have previously been reported in studies following individuals from midlife or younger old ages[7 8] but not among the oldest

BMJ

old.[8] [9] In a Norwegian health examination study, an educational pattern was present only among cohorts aged below 70 years at baseline but not among those aged 70 years and over.[8] Similarly, among a Finnish cohort aged 90 years and over, no statistically significant educational gradient in dementia mortality emerged.[9] The lack of educational differentials among the oldest old may relate to selective survival. People with lower education experience higher mortality at younger ages, and those who survive to older ages do so because of their better health. Thus, the population surviving to older ages is more homogeneous in terms of health-related characteristics and, as a result, the socioeconomic differences in mortality are diminished. Another possible explanation for the lack of educational gradient in dementia mortality is the fact that the distribution of education in the oldest cohorts is highly skewed. Given that the majority of people in these cohorts have no more than basic education, other indicators of socioeconomic position (SEP) may be more suitable for identifying high-risk population subgroups.[2] [10] Previous studies suggest that among adults in general, overall mortality disparities are greater or have increased to a greater extent in terms of occupational social class[11] and income[12] [13] than education. Among the Finnish cohort of nonagenarians,[9] occupational social class was a strong predictor of dementia mortality with a threefold hazard of dementia death among the unskilled manual workers compared with upper non-manuals. Personal income in midlife, however, was not related to dementia mortality among a cohort of Norwegian men.[14] To our knowledge, no previous study has assessed inequalities in dementia mortality by household income, a socioeconomic indicator that is more directly related to material resources available to the individual and that more rigorously captures the living conditions of the most disadvantaged population subgroups. A low household income may, in addition to material disadvantage, induce psychosocial stress, increasing the risk of dementia directly or through less favourable health behaviours. Disentangling the contributions of education, occupational social class and household income will thus provide important insights into the potential mechanisms how SEP shapes the risk of dementia death.

This study contributes to the existing knowledge by assessing socioeconomic inequalities in dementia mortality using multiple indicators of SEP, including education, occupational social class and household income. More specifically, the aims of the study were to (1) investigate the magnitude of socioeconomic inequalities in dementia mortality in relation to age, and compare the patterns to those in mortality from all other causes of death, (2) to quantify the contribution of dementia to the socioeconomic inequalities in overall mortality at older ages and (3) to assess whether education, occupational social class and household income are independently related to dementia mortality once the other indicators are taken into account. This was because different indicators of SEP are correlated but each of them may

have independent associations with dementia mortality. We further estimated models adjusting for confounders including marital status and chronic health conditions. We used longitudinal registry data on a large population-based sample, which permit a 15-year follow-up of dementia-related deaths with no attrition or recall bias. All indicators of SEP were measured in midlife in order to avoid selection to socioeconomic groups on the basis of cognitive decline.

## METHODS
### Sample
We used an 11% random sample of the Finnish population in 1987–2007 drawn from the Statistics Finland population register, which covers all permanent residents. Statistics Finland linked the sample with information from various administrative registers including the National Death Register and healthcare registers using unique personal identification numbers assigned to all permanent residents.

In the present study, we included men and women aged 70–87 years at the end of year 2000. For these cohorts, midlife socioeconomic characteristics could be identified using information from the population censuses conducted in 1970, 1975, 1980 and 1985. Individuals with missing census information due to residing outside of Finland (n=920) and those with missing household income information due to not being part of the household population in the census year (n=401) were excluded. Seven individuals emigrated during the first year of follow-up and thus were excluded from the analyses. The analytic sample consisted of 54 964 individuals.

### Mortality data
Dates and causes of death were obtained from the Death Register. Dementia-related deaths were identified using the International Classification of Diseases 10th revision codes F00–03 and G30 as the underlying or any of the three contributory causes of death reported on the death certificate. We identified 11 395 persons who died from dementia and 30 637 persons who died from other causes during the follow-up in 2001–2016.

### Indicators of SEP
The information of all indicators of SEP was derived from the quinquennial population censuses of 1970–1985. A particular census year was chosen on the basis of the study subject's age so that the indicators were measured at around the age of 55 years (range 53–57 years) for all. Education was indicated as the highest achieved qualification, categorised as tertiary (generally 13+ years of education; International Standard Classification of Education (ISCED)-1997 codes 5–6), secondary (10–12 years, ISCED 3–4) and basic education/no qualifications (9 years, ISCED 0–2). Occupational social class comprised five groups, classified as non-manual, manual, self-employed farmer, other self-employed and no occupation/unknown. Information of occupational

social class in the census year was lacking for 10 465 individuals due to non-employment at that time. For 9942 individuals, the information could nevertheless be obtained from previous years in which the individuals were employed. Household income indicated the taxable annual income of all household members, including all income received in money or monetary benefit subject to tax. The information was obtained from the Finnish Tax Administration and the Social Insurance Institution of Finland. We adjusted for household composition using the Organisation for Economic Co-operation and Development(OECD) modified equivalence scale.[15] Income quintiles were formed based on the household income distribution in the population aged 15 years and over in the census year.

## Covariates

The analyses incorporated information of economic activity measured from the census year because being out of the labour market may indicate poor health and affect dementia risk independently but also lead to reduced household income. Economic activity was classified as being in the labour force, retired and other inactive. Marital status was measured at baseline (the end of 2000), classified as married, divorced, widowed and never married. Baseline chronic health conditions included indicators of vascular and lifestyle risk factors for dementia,[16] and were identified from health registers in the 5-year period before the baseline, covering 1996–2000. We used the diagnostic records of the hospital discharge register and patient censuses of the National Institute for Health and Welfare, and the records of prescription medicine purchases and of entitlement to special reimbursement for the medication expenses for certain chronic diseases maintained by the Social Insurance Institution of Finland. We included indicators for alcohol-related diseases and accidental poisoning by alcohol, asthma and other chronic obstructive pulmonary disease (COPD), diabetes and heart disease (for coding see online supplementary table 1). These chronic conditions may confound the association between midlife SEP and dementia mortality as the diseases usually develop over a long period of time and thus reflect health behaviours or health problems already present in midlife. To account for potential regional variance in socioeconomic characteristics and mortality, we included dummies for region of residence (Western Finland, Helsinki capital region, rest of Southern Finland, Eastern Finland and Lapland) and the degree of urbanisation of the municipality of residence, a variable based on the proportion of population living in urban settlements and the population of the largest urban settlement in the municipality (urban, semiurban and rural).

## Statistical analyses

We followed the study population for dementia mortality from 1 January 2001 until 31 December 2016. Individuals were censored on the date of death, at the end of the year preceding emigration, or at the end of 2016, whichever came first.

For descriptive statistics, we calculated age-adjusted dementia mortality rates per 10 000 person-years at risk by indicators of SEP and the covariates. In order to assess the magnitude of socioeconomic inequalities in relation to age, we estimated Kaplan-Meier survival functions by education, occupational social class and household income. In these analyses, we contrasted the survival functions of the highest and lowest education groups, non-manual and manual employees, and the highest and lowest household income quintiles. The equality of survival functions was tested using log-rank tests. For the comparison between dementia mortality and the more general mortality patterns, separate Kaplan-Meier survival functions were estimated for mortality from all other causes of death. We also estimated HRs and their 95% CIs for low versus high socioeconomic groups at the age of 70–79, 80–89 and 90 years and over.

To quantify the contribution of dementia to socioeconomic differences in overall mortality at older ages, we calculated absolute rate differences in mortality between socioeconomic groups (basic vs tertiary education, manual vs non-manual occupational social class, lowest vs highest household income quintile) by cause of death. The contribution was determined by the rate difference in dementia mortality as a percentage of the rate difference in total mortality. Because the level of dementia mortality increases substantially with age, we also assessed age-specific contributions (at the age of 70–79, 80–89 and 90+ years).

To estimate the independent associations between each indicator of SEP and dementia mortality, we used Cox regression models. Attained age in years was used as the time scale, and thus all analyses adjusted for the confounding effect of age.[17] We first estimated crude associations between each indicator and dementia mortality, adjusting for calendar year dummies, gender, region of residence and the degree of urbanisation (model 1). Model 2 included education, occupational social class and household income as covariates, thus showing mutually adjusted associations. Midlife economic activity was adjusted for in model 3. We further adjusted for baseline marital status and chronic health conditions in model 4 to assess the extent to which these confounding factors attenuated the relative hazard attached to each socioeconomic indicator.

We tested for interactions between gender and each socioeconomic indicator using likelihood ratio test. Interactions were statistically non-significant (p>0.05), and thus we conducted all analyses for men and women combined. We also tested for interactions of all pairwise combinations of the socioeconomic indicators, adjusting for the covariates of model 1. These interactions were all statistically non-significant (p>0.05). All analyses were performed using Stata V.15.1.[18]

## Patient and public involvement

No patients were involved in setting the research question or the outcome measures, nor were they involved

## RESULTS

Table 1 shows the distribution of the study population by indicators of midlife SEP, economic activity and baseline characteristics. The vast majority of individuals (77.2%) had no higher than basic education, and manual employees formed the largest occupational social class (43.6%). Higher household income quintiles were over-represented among the study population due to the higher incomes of the middle aged compared with the rest of the population and also partly because of greater mortality of the lower income groups between the time of measurement of midlife income and the baseline. During the 528 387 person-years at risk 11 395 individuals died from dementia, the average age-adjusted dementia mortality rate being 223.1 and 210.8 per 10 000 person-years among men and women, respectively. The rate was higher for those with lower education, occupational social class and household income, and also for the non-married and people with chronic health conditions apart from asthma and other COPD.

Kaplan-Meier survival functions in figure 1 show that dementia mortality differed by all indicators of SEP (log rank test, p<0.001 for each indicator), and that the age patterns differed between the indicators (for 95% CIs see online supplementary table 2). The inequalities emerged at an earlier age when SEP was measured in terms of household income (panel C) compared with education (panel A) and occupational social class (panel B). At the age of 90 years and above, by contrast, the differences were more pronounced when SEP was measured in terms of education. Nevertheless, inequalities in dementia mortality emerged substantially later in life compared with mortality from all other causes. HRs in table 2 show that relative inequalities in mortality tended to diminish with age for all indicators of SEP regardless of cause of death. However, education differences in dementia mortality showed a different age pattern in that the point estimates indicated stable inequality with age.

Overall, dementia contributed to 28.1% of educational and 20.9% of household income differences in total mortality at the age of 70 years and over (table 2). The contribution to occupational social class differences was somewhat smaller (16.7%). The contribution of dementia to socioeconomic inequalities substantially increased from the age of 70–79 to 90 years and over.

Cox regression models in table 3 show adjusted HRs for dementia mortality across all ages from 70 years and over. Adjusted for calendar year, gender, region of residence and the degree of urbanisation in model 1, the associations were strongest for basic education (HR=1.23, 95% CI 1.15 to 1.32), unknown occupational social class

(HR=1.20, 1.00 to 1.44) and the lowest household income quintile (HR=1.28, 1.20 to 1.35). Mutual adjustment of socioeconomic indicators in model 2 attenuated educational differences by about 40%, and unknown occupational social class to a non-significant level. Basic education (HR=1.14, 1.06 to 1.23), manual occupational social class (HR=1.06, 1.01 to 1.11) and three lowest household income quintiles (for the lowest quintile HR=1.24, 1.16 to 1.32) all predicted dementia mortality independently of each other. Adjustment for midlife economic activity in model 3 attenuated the excess hazard particularly of the lower household income quintiles. Adjustment for baseline marital status and chronic health conditions in model 4 contributed to a small change in the estimates, the attenuation being largest for the lowest household income quintile. In this full model, basic education increased the hazard of dementia death by 14% (1.06 to 1.23), manual occupational social class by 5% (1.00 to 1.10) and the two lowest household income quintiles by 7%–13% (HR=1.07, 1.00 to 1.14 to HR=1.13, 1.06 to 1.22).

## DISCUSSION
### Main findings and their interpretation

In this study we have shown that dementia mortality at older ages is socioeconomically patterned in terms of multiple indicators of SEP. People with lower education, occupational social class and household income have a higher risk of dementia death compared with those with higher SEP. These results add to the literature on socioeconomic inequalities in old-age mortality, which has previously shown a socioeconomic pattern in many other specific causes of death such as cardiovascular diseases, COPD and cancer.[1] Our results indicate, moreover, that dementia is an important factor in overall socioeconomic inequalities in old-age mortality, contributing to 21%–28% of household income and educational differences in total mortality among the population aged 70 years and over. The contribution of dementia to overall socioeconomic inequalities in mortality increased substantially with age, which relates to the increasing proportion of deaths attributable to dementia with advancing age.[19]

A major difference in the patterns between dementia mortality and mortality from all other causes of death was that socioeconomic inequalities in dementia mortality emerged later and the inequalities in dementia mortality between high and low education groups persisted in the same magnitude to the oldest old ages (90 years and above). By contrast, inequalities in mortality from other causes of death tended to diminish with age. The attenuation of socioeconomic inequalities with age is a general finding,[1 2] and may partly relate to selective survival, suggesting that people who survive to very old age have more similar health profiles across socioeconomic groups. Our results show, however, that even among people who survive to the oldest old age, education groups differ in neurological health. This is a novel finding in that

**Table 1** Distribution of the study population, dementia deaths and age-adjusted dementia mortality rates (per 10 000 person-years) by indicators of midlife socioeconomic position and economic activity and baseline characteristics, Finnish men and women in 2001–2016

| | N (%) | Dementia deaths | | |
| | | n | Rate | 95% CI |
|---|---|---|---|---|
| Mean age at baseline (SD) | 76.4 (4.8) | | | |
| Gender | | | | |
| Men | 20 100 (36.6) | 3409 | 223.1 | 215.6 to 230.5 |
| Women | 34 864 (63.4) | 7986 | 210.8 | 206.3 to 215.4 |
| Education* | | | | |
| Tertiary | 5445 (9.9) | 1014 | 185.5 | 174.3 to 196.7 |
| Secondary | 7074 (12.9) | 1446 | 205.7 | 195.4 to 216.1 |
| Basic | 42 445 (77.2) | 8936 | 221.8 | 217.3 to 226.3 |
| Occupational social class* | | | | |
| Non-manual | 17 015 (31.0) | 3524 | 201.1 | 194.6 to 207.6 |
| Manual | 23 951 (43.6) | 4882 | 228.4 | 222.1 to 234.6 |
| Self-employed farmer | 10 204 (18.6) | 2211 | 215.7 | 206.9 to 224.6 |
| Other self-employed | 3271 (6.0) | 657 | 212.2 | 196.3 to 228.1 |
| No occupation/unknown | 523 (1.0) | 121 | 239.2 | 196.5 to 282.0 |
| Household income* | | | | |
| Highest quintile | 13 667 (24.9) | 2715 | 196.9 | 189.7 to 204.2 |
| Second | 10 522 (19.1) | 2098 | 209.6 | 200.9 to 218.4 |
| Third | 10 110 (18.4) | 2114 | 217.3 | 208.2 to 226.3 |
| Fourth | 10 292 (18.7) | 2183 | 223.2 | 214.0 to 232.3 |
| Lowest quintile | 10 373 (18.9) | 2285 | 241.5 | 231.8 to 251.2 |
| Economic activity* | | | | |
| Active | 37 266 (67.8) | 7585 | 208.1 | 203.6 to 212.7 |
| Retired | 8881 (16.2) | 1742 | 257.1 | 245.2 to 269.0 |
| Other inactive | 8817 (16.0) | 2068 | 212.7 | 203.7 to 221.8 |
| Marital status | | | | |
| Married | 24 789 (45.1) | 4471 | 208.7 | 202.6 to 214.8 |
| Divorced | 4056 (7.4) | 797 | 237.0 | 220.9 to 253.2 |
| Widowed | 20 997 (38.2) | 5000 | 214.3 | 208.4 to 220.3 |
| Never married | 5122 (9.3) | 1127 | 240.9 | 227.2 to 254.7 |
| Chronic health conditions | | | | |
| Alcohol-related diseases | 308 (0.6) | 68 | 505.3 | 381.2 to 629.3 |
| Asthma and COPD | 4510 (8.2) | 789 | 232.9 | 216.9 to 248.9 |
| Diabetes | 6714 (12.2) | 1240 | 275.0 | 259.8 to 290.2 |
| Heart disease | 18 094 (32.9) | 3562 | 237.1 | 229.5 to 244.7 |
| Region of residence | | | | |
| Western Finland | 25 078 (45.6) | 4979 | 204.1 | 198.6 to 209.7 |
| Helsinki capital region | 7449 (13.6) | 1582 | 208.3 | 198.3 to 218.4 |
| Rest of Southern Finland | 12 056 (21.9) | 2464 | 214.8 | 206.5 to 223.0 |
| Eastern Finland | 8458 (15.4) | 1916 | 250.1 | 239.2 to 261.0 |
| Lapland | 1923 (3.5) | 454 | 261.5 | 238.1 to 285.0 |
| Degree of urbanisation | | | | |
| Urban | 29 853 (51.0) | 6401 | 217.2 | 212.0 to 222.4 |

**Table 1** Continued

| | | Dementia deaths | | |
| | N (%) | n | Rate | 95% CI |
| --- | --- | --- | --- | --- |
| Semiurban | 9285 (17.7) | 1831 | 210.6 | 201.2 to 220.0 |
| Rural | 15 826 (31.3) | 3163 | 215.3 | 208.0 to 222.7 |
| Total | 54 964 (100.0) | 11 395 | 215.7 | 211.7 to 219.7 |

*Information from the population censuses of 1970–1985, the study population being aged 53–57 years.
COPD, chronic obstructive pulmonary disease.

previous studies have identified consistent socioeconomic inequalities in dementia mortality only among the younger old[7 8] but the results have been mixed for the oldest old.[8 9] Participation bias may at least partly explain the differences in findings; people of older age, lower SEP and with health problems are less likely to participate in surveys and studies involving health examinations. Our study employed register data on a population-based cohort and thus is not affected by participation or attrition biases.

The age patterns in dementia mortality differed between indicators of SEP: while educational differences were more pronounced among the oldest old (90 years and over), the differences among the younger old (70–79 years) were largest when SEP was measured in terms of household income. Individuals in the lowest income quintile represent the most disadvantaged population subgroups with multiple potential dementia risk factors. Our findings show that the higher dementia mortality of the lowest household income quintiles was strongly—although not fully—confounded by greater morbidity of these groups. Severe health problems that were already present in midlife have potentially affected both household incomes and the risk of dementia death. However, we cannot rule out the possibility of mediation, especially because chronic health conditions were measured after midlife income; impoverished material conditions may also affect dementia risk through, for example, health-related behaviours, cardiovascular risk factors[16] and psychological stress.[20] In the presence of mediation, our estimates would be conservative as they would overadjust part of the effect of socioeconomic disadvantage. Future studies are needed to establish the causal relationship between mediating factors and dementia mortality using mediation analysis techniques.

Education, in turn, may have particular benefits above and beyond physical health factors among the population surviving to the oldest old age. Our results show persistent educational differences in dementia mortality, and the association was not confounded by chronic health conditions, economic activity or marital status. Education is a well-established predictor of dementia incidence,[21] although the exact mechanisms are still not known. Brain autopsy studies indicate, in line with the cognitive reserve hypothesis,[22] that education is not associated with the burden of neuropathology at death but higher education enables individuals to compensate longer for the neuropathological changes before developing clinical symptoms of dementia.[23] Thus, it is possible that the educational differences in dementia mortality we found in our study are due to competing risks; people with higher education died from other causes before they reached the phase of clinical dementia or died from other causes before dementia progressed to death. However, the empirical evidence for the cognitive reserve hypothesis remains open to debate. For example, several studies have not identified educational differences in survival time after dementia onset,[24] which is among the key hypotheses in the cognitive reserve model.[25] Therefore, it is plausible that higher education enhances brain health and protects against (or postpones) not only the clinical symptoms but also the development of neurodegenerative disorders.

Occupational social class differences in dementia mortality were modest following adjustment for education and household income. In particular, the high hazard among those with no occupation disappeared after these adjustments indicating that this group experienced multiple socioeconomic disadvantages. The results suggest, nevertheless, that higher social class occupations may involve greater cognitive demands and intellectual engagement, and thus enhance cognitive health.[26 27] In contrast, lower class occupations or long periods of economic inactivity due to unemployment or early retirement may reduce opportunities for cognitive investment. Overall, the results of this study suggest that all three indicators of SEP are important factors in bringing about socioeconomic differences in dementia mortality, also influencing inequalities in overall mortality among the older population.

### Methodological considerations

We used a unique population-representative sample of older adults in Finland with 11 395 dementia deaths identified from the National Death Register. The register-based sample was not affected by participation or attrition bias, which are common limitations of many cohort designs, particularly among the older population. The population register encompasses rich information on demographic and socioeconomic characteristics of individuals over the life course, and is not subject to bias from individuals' self-reports or recollection.

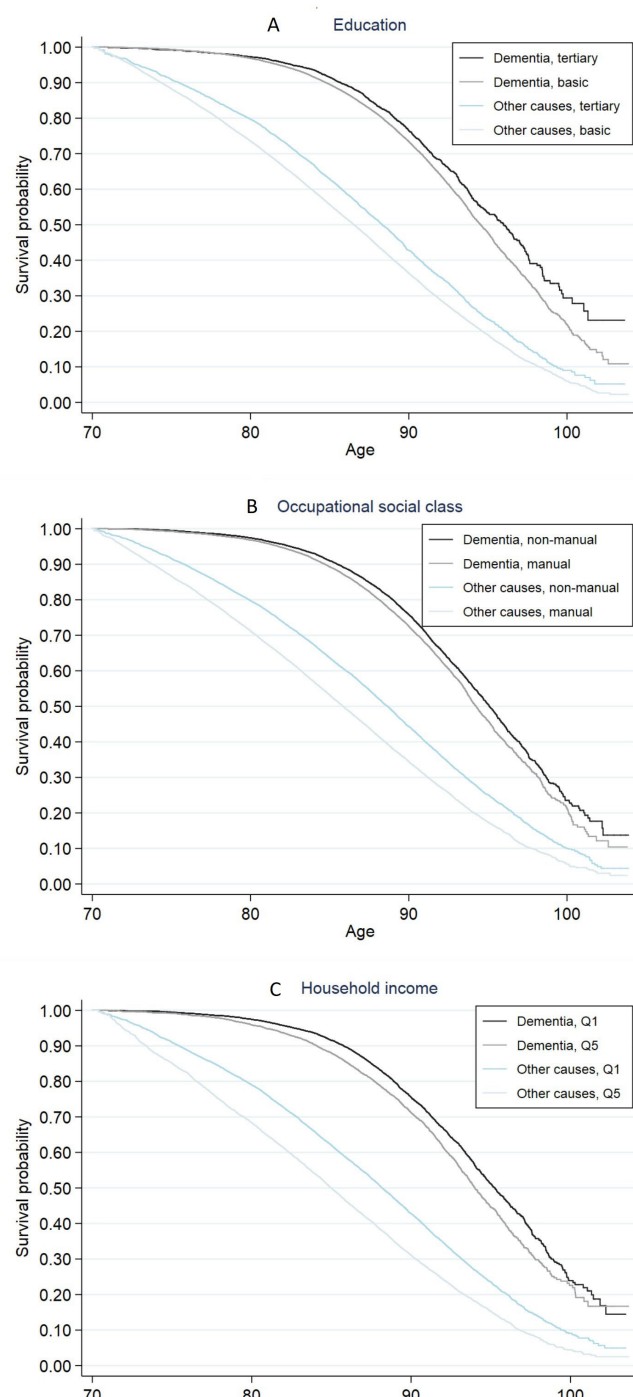

**Figure 1** Kaplan-Meier survival probabilities for dementia mortality and mortality from all other causes of death by (A) education, (B) occupational social class and (C) household income quintile (Q1=highest, Q5=lowest), Finnish men and women in 2001–2016. Information of midlife socioeconomic position obtained from the population censuses of 1970–1985, the study population being aged 53–57 years.

dementia as the cause of death has improved since the late 1990s, and the specificity is particularly high.[28] To minimise any bias arising from potential under-reporting of dementia as the underlying cause of death, we applied the multiple-cause approach and included also cases where dementia was recorded as any of the three contributory causes.[29] Defined this way, we identified 21% of all deaths at the age of 70 years and over to be attributable to dementia. This relatively high proportion is in line with that reported in England and Wales, where dementia accounted for 19% of all deaths at the age of 80 years and over.[30] Furthermore, we ran sensitivity analyses with interaction with calendar year, and found that the associations between the indicators of SEP and dementia mortality did not vary in time. Therefore, we believe our results are not biased by over-reporting or under-reporting of dementia as the cause of death or by changes in documentation practices.

Second, the information of household income was based on taxable income and the variable thus excludes certain monetary transfers such as housing allowance and social assistance. These means-tested sources of income may be especially relevant for people with health problems and those outside the labour market. This might lead to overestimation of the income effect. Information of disposable income was not available for years 1970–1985, but we carried out a robustness check for the correlation between taxable and disposable household incomes (as continuous variables) using the population aged 15 years and over in 1995 and found the correlation to be as high as 0.97 (among the population aged 53–57 years in 1995 the correlation was 0.98). Therefore, it is unlikely that the use of disposable income would change the ranking of individuals in the household income distribution to the extent that it would affect our main findings.

Finally, the causal relationship between SEP and dementia is difficult to establish in observational studies. We therefore measured all socioeconomic characteristics 15–30 years before the mortality follow-up, and it is thus very unlikely that any symptoms of dementia affected the midlife socioeconomic attainment of individuals. Nevertheless, we cannot exclude the possibility that early cognitive decline may have affected midlife SEP, especially measured in terms of occupational social class and household income. Also, given the small proportion of people with tertiary education in these cohorts (10%), it is possible that this forms a select group with multiple advantages including higher childhood SEP and early cognitive ability. Because register data do not cover information of traditional risk factors related to health behaviours such as smoking and physical activity, we included indicators of chronic conditions to measure cardiovascular and life style risk factors for dementia that may confound the association between SEP and dementia mortality.

Despite the rich register data, our study also has some limitations. First, we could only identify dementia cases that have been recorded on the death certificate. According to a validation study for identifying dementia in the Finnish national registers, the documentation of

**Table 2** Relative and absolute differences in mortality between high and low socioeconomic groups* by cause of death and age, and contribution (%) of dementia and other causes of death to socioeconomic differences in total mortality by age, Finnish men and women in 2001–2016

| | 70–79 years | | | | 80–89 years | | | | 90+ years | | | | All ages 70+ years |
| | HR | 95% CI | Rate difference | Contribution (%) | HR | 95% CI | Rate difference | Contribution (%) | HR | 95% CI | Rate difference | Contribution (%) | Contribution (%) |
|---|---|---|---|---|---|---|---|---|---|---|---|---|---|
| **Education†** | | | | | | | | | | | | | |
| Dementia | 1.24 | 0.97 to 1.58 | 8.7 | 8.3 | 1.19 | 1.09 to 1.29 | 40.4 | 33.5 | 1.24 | 1.10 to 1.40 | 157.9 | 51.8 | 28.1 |
| Other causes | 1.38 | 1.26 to 1.50 | 96.1 | 91.7 | 1.11 | 1.06 to 1.17 | 83.0 | 68.9 | 1.13 | 1.03 to 1.24 | 146.9 | 48.2 | 71.8 |
| Total mortality | 1.36 | 1.25 to 1.48 | 104.8 | 100.0 | 1.13 | 1.09 to 1.18 | 120.4 | 102.4 | 1.17 | 1.09 to 1.26 | 304.8 | 100.0 | 100.0 |
| **Occupational social class‡** | | | | | | | | | | | | | |
| Dementia | 1.22 | 1.03 to 1.44 | 7.7 | 6.3 | 1.17 | 1.10 to 1.23 | 35.3 | 20.3 | 1.09 | 1.00 to 1.17 | 63.5 | 22.8 | 16.7 |
| Other causes | 1.44 | 1.36 to 1.53 | 114.2 | 93.8 | 1.24 | 1.20 to 1.29 | 138.7 | 79.7 | 1.19 | 1.12 to 1.27 | 215.0 | 77.2 | 83.3 |
| Total mortality | 1.41 | 1.34 to 1.49 | 121.8 | 100.1 | 1.22 | 1.19 to 1.26 | 173.9 | 100.0 | 1.15 | 1.09 to 1.21 | 278.4 | 100.0 | 100.0 |
| **Household income§** | | | | | | | | | | | | | |
| Dementia | 1.63 | 1.32 to 2.01 | 22.4 | 12.9 | 1.22 | 1.14 to 1.31 | 46.3 | 21.4 | 1.19 | 1.08 to 1.32 | 135.6 | 35.2 | 20.9 |
| Other causes | 1.54 | 1.43 to 1.66 | 151.0 | 87.1 | 1.24 | 1.19 to 1.30 | 169.7 | 78.6 | 1.19 | 1.10 to 1.28 | 249.1 | 64.8 | 79.2 |
| Total mortality | 1.55 | 1.44 to 1.67 | 173.3 | 100.0 | 1.24 | 1.19 to 1.29 | 216.0 | 100.0 | 1.19 | 1.12 to 1.26 | 384.7 | 100.0 | 100.0 |

HRs adjusted for calendar year. Age-adjusted incidence rates calculated as dementia deaths per 10 000 person-years at risk. Contribution of dementia determined by the rate difference in dementia mortality as a percentage of the rate difference in total mortality.

*Information from the population censuses of 1970–1985, the study population being aged 53–57 years.

†Tertiary versus basic education.

‡Non-manual versus manual occupational social class.

§Highest versus lowest household income quintiles.

**Table 3** HRs and 95% CIs for dementia mortality by indicators of midlife socioeconomic position,* Finnish men and women in 2001–2016, n=54 964

| Indicator of socioeconomic position | Model 1† | | Model 2‡ | | Model 3§ | | Model 4¶ | |
|---|---|---|---|---|---|---|---|---|
| | HR | 95% CI | HR | 95% CI | HR | 95% CI | HR | 95% CI |
| Education | | | | | | | | |
| Tertiary | 1.00 | | 1.00 | | 1.00 | | 1.00 | |
| Secondary | 1.14 | 1.05 to 1.23 | 1.08 | 0.99 to 1.17 | 1.08 | 0.99 to 1.17 | 1.08 | 0.99 to 1.18 |
| Basic | 1.23 | 1.15 to 1.32 | 1.14 | 1.06 to 1.23 | 1.14 | 1.05 to 1.22 | 1.14 | 1.06 to 1.23 |
| Occupational social class | | | | | | | | |
| Non-manual | 1.00 | | 1.00 | | 1.00 | | 1.00 | |
| Manual | 1.14 | 1.09 to 1.20 | 1.06 | 1.01 to 1.11 | 1.05 | 1.00 to 1.11 | 1.05 | 1.00 to 1.10 |
| Farmer | 1.08 | 1.02 to 1.15 | 0.96 | 0.90 to 1.03 | 0.97 | 0.91 to 1.04 | 0.98 | 0.92 to 1.05 |
| Other self-employed | 1.05 | 0.96 to 1.14 | 0.98 | 0.90 to 1.07 | 0.99 | 0.91 to 1.08 | 1.00 | 0.92 to 1.09 |
| No occupation/ unknown | 1.20 | 1.00 to 1.44 | 1.04 | 0.87 to 1.25 | 0.94 | 0.78 to 1.14 | 0.94 | 0.78 to 1.13 |
| Household income | | | | | | | | |
| Highest quintile | 1.00 | | 1.00 | | 1.00 | | 1.00 | |
| Second | 1.08 | 1.02 to 1.14 | 1.04 | 0.98 to 1.10 | 1.03 | 0.98 to 1.10 | 1.02 | 0.96 to 1.09 |
| Third | 1.13 | 1.07 to 1.20 | 1.08 | 1.02 to 1.15 | 1.07 | 1.00 to 1.14 | 1.05 | 0.99 to 1.12 |
| Fourth | 1.17 | 1.10 to 1.24 | 1.13 | 1.06 to 1.20 | 1.10 | 1.03 to 1.17 | 1.07 | 1.00 to 1.14 |
| Lowest quintile | 1.28 | 1.20 to 1.35 | 1.24 | 1.16 to 1.32 | 1.18 | 1.10 to 1.26 | 1.13 | 1.06 to 1.22 |

All models used age as time scale and adjusted for calendar year, gender, region of residence and the degree of urbanisation.
*Information from the population censuses of 1970–1985, the study population being aged 53–57 years.
†Model 1: each indicator of socioeconomic position separately.
‡Model 2: indicators of socioeconomic position mutually adjusted.
§Model 3: model 2+midlife economic activity.
¶Model 4: model 3+baseline marital status and chronic health conditions (alcohol-related diseases, asthma and chronic obstructive pulmonary disease, diabetes and heart disease).

## CONCLUSIONS

This study provides new insight into the socioeconomic inequalities in old-age mortality by showing a consistent pattern in dementia mortality by multiple indicators of SEP. Low education, occupational social class and household income were all associated with higher risk of dementia death, although the socioeconomic differences emerged later than in mortality from other causes. Household income differences in dementia mortality were more pronounced among the younger old, and the associations were largely attributable to other chronic health conditions such as diabetes and alcohol-related diseases. Educational inequalities, by contrast, were independent of chronic health conditions and became more pronounced at the oldest old age where mortality inequalities generally begin to attenuate. The results indicate that dementia mortality may be amenable to socioeconomic interventions in midlife. The findings also suggest that dementia contributes to socioeconomic inequalities in overall mortality at older ages and, thus, dementia prevention is important from the point of view of socioeconomic inequalities in total mortality.

**Contributors** KK, EE, TL, LT and PM participated in designing the study, generating hypotheses, interpreting the data and critically revised the manuscript for important intellectual content. KK analysed the data, conducted the literature review and wrote the first draft of the manuscript.

**Funding** KK was supported by the Eino Jutikkala Fund. PM was supported by the Academy of Finland and the Strategic Research Council PROMEQ project (#303615). PM was also supported by the European Union Horizon2020 Programme under grant agreement no. 667661 (promoting mental wellbeing in the ageing population—MINDMAP).

**Disclaimer** The study does not necessarily reflect the Commission's views and in no way anticipates the Commission's future policy in this area. The funders had no role in study design, data collection and analysis, decision to publish or preparation of the manuscript.

**Competing interests** None declared.

**Patient consent for publication** Not required.

**Ethics approval** The study has been approved by Statistics Finland Board of Ethics (permit TK-53-339-13). The data were collected for routine administrative registration purposes and, therefore, informed consent of the participants was not obtained. These register data can be used for scientific purposes under the Personal Data Act and the Statistics Act. Statistics Finland anonymised the data prior to providing them to researchers.

**Provenance and peer review** Not commissioned; externally peer reviewed.

**Data availability statement** Data may be obtained from a third party and are not publicly available. Statistics Finland, the National Institute for Health and Welfare and the Social Insurance Institution of Finland have collected and own the data. Due to data protection regulations, the authors are not allowed to make the data available to third parties. Researchers can apply for data access by contacting the register-holding institutions: Statistics Finland (http://stat.fi/index_en.html); National

Institute for Health and Welfare (https://thl.fi/en); Social Insurance Institution of Finland (https://www.kela.fi/web/en).

**ORCID iD**
Kaarina Korhonen http://orcid.org/0000-0001-8499-2008

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
