## [Reviewer comments · BMJ Open]

ARTICLE DETAILS

TITLE (PROVISIONAL)	Midlife socioeconomic position and old-age dementia mortality: a large prospective register-based study from Finland
AUTHORS	Korhonen, Kaarina; Einiö, Elina; Leinonen, Taina; Tarkiainen, Lasse; Martikainen, Pekka

VERSION 1 - REVIEW

REVIEWER	Almar Kok Amsterdam UMC, location VUmc, department of Epidemiology & Biostatistics, Amsterdam, The Netherlands
REVIEW RETURNED	12-Aug-2019

GENERAL COMMENTS	“Midlife socioeconomic position and old-age dementia mortality: a large prospective register-based study from Finland” This paper describes an analysis of Finnish register data, to determine socioeconomic inequalities – measured by education, occupation and income, in dementia incidence. The paper is well written and relevant. The analyses seem generally appropriate. Although the manuscript is already strong, I have listed a number of instances in which clarification or amendments may be needed. (in order of the manuscript) #1. Page 4 line 36-39. Although the argument of the authors is correct (and is mentioned multiple times throughout the manuscript), I feel the authors should explicitly mention the crucial role of SEP in the selective survival effect. Those with a lower SEP tend to die earlier, and this is the reason that the surviving sample is more homogeneous in terms of education. Additionally, from the lower SEP groups, only the very healthiest survive to high ages, leading to a decrease in observed socioeconomic inequalities. This is often referred to as the “age-as-leveler” hypothesis. #2. I consider the disentangling of the contributions of education, occupation and income to dementia mortality as an important strength of this study. However it would benefit the introduction if the authors could elaborate on why this is important in the context of dementia; which different pathways are they thinking about with regards to dementia specifically? E.g., I can think of some specific links with education and with occupation (e.g., more complex work), but for income this is less clear. What effects would income have on dementia independent from education and occupation? The general argument on p.5, lines 3-8 is, in my opinion, not specific enough.
--

#3. Why would the authors want to capture mediating pathways? It seems to be beyond the focus of this study to examine whether chronic health conditions and marital status would mediate the effect of SEP on dementia mortality. Moreover, no formal mediation analysis is performed. It is certainly important to adjust for confounding factors, and marital status might be a confounder because household rather than individual income is measured. But I don't see the added value of "adjusting for" chronic health conditions here. The authors might still regard them as potential confounders (perhaps because they suspect health selection effects), but then they must conceptualize them as such.

#4. Page 6 line 22-24. At first glance, the proportion of 20.7% of dementia deaths relative to all deaths seems very high. Could the authors elaborate more (either in the methods or discussion) on this number? Perhaps there are reasons for this (e.g., they counted primary, secondary and tertiary cause of death), but it is currently not clear how this incidence relates to incidences from other major studies.

#5. Page 6, line 42. The authors state that there is also an education category "unknown". But this category does not appear again in the manuscript, also not in the Tables. Where did it go?

#6. Page 6, line 46. How many individuals had no information on occupational social class at baseline?

#7. Page 6, line 51-53. The fact that no information on tax-free income transfers is included may be a major limitation, because it could lead to overestimation of socioeconomic inequalities. This should at least be discussed.

#8. Page 8. It may be relevant to not only work with rate differences (absolute differences in death rates) but also rate ratios (relative differences in death rates). This might provide important information, e.g. in terms of the development of inequalities across age groups. This could also better qualify the authors' statements about how the inequalities in dementia mortality compare to inequalities in overall mortality. I am not sure whether their present comparison takes into account that the dementia rates are generally much lower than overall mortality rates.

#9. Page 8. I am not sure if the method used by the authors to calculate the contribution of dementia to socioeconomic inequalities to overall mortality is the same or intended to do the same as calculating the Population Attributable Fraction (PAF)? If so, this could be mentioned in the manuscript.

#10. Page 10 / Table 2. The figure that about 66% of socioeconomic inequalities in mortality above age 70 can be attributed to dementia deaths seems quite high to me (29.9% + 12.0% + 24.0%). Were the SEP indicators mutually adjusted here? Resonating my comment #4, are these surprising or unsurprising numbers in light of previous studies?

#11. Page 13, line 21. It is not clear what the authors mean with education "truly" enhancing brain health. Clarification and references could help here.

#12. Page 13. Please discuss the high dementia death rate in the "unknown" occupation category.

#13. Discussion. It would be valuable if the implications of finding that different socioeconomic indicators are independently related to dementia mortality are discussed. What different pathways could they indicate? What does this mean for prevention or intervention?

#14. Table 1. For clarification, please add to the Table title that SEP was measured at about age 55.

	#15. Table 3. Table title says Table 2, should be 3.
--	--

REVIEWER	Linda Enroth Tampere University, Finland
REVIEW RETURNED	19-Aug-2019

GENERAL COMMENTS	Thank you for inviting me to review the manuscript "Midlife socioeconomic position and old-age dementia mortality: a large prospective register-based study from Finland". The study aimed to examine the association of three socioeconomic position indicators with dementia-related mortality and the contribution of dementia to overall mortality differences at older ages. The study showed that in the Finnish older population, all three socioeconomic position indicators were independently associated with dementia mortality (higher dementia mortality in lower socioeconomic positions), dementia accounted 25-30% of the total mortality between high and low educated and according to household income but less according to occupational social class in 70+ population, and that the contribution of dementia on socioeconomic mortality differences increased with age. This study adds to the current knowledge that dementia is an important contributor of socioeconomic inequality in old age mortality. Regarding the rapid increase in the oldest old population in Finland and globally, this study is timely and provides new information as well as strengthen already reported socioeconomic inequality in dementia mortality in the oldest old in Finland. The study is well conducted however, I have few comments and questions for clarifications for the authors. General comments:  1. The study uses three socioeconomic position indicators namely education, occupational social class and household income, which all have more than two categories. Could authors provide an explanation why analyses were conducted contrasting only, so to say, extreme ends? Authors elaborate in the discussion section on the found differences in results based on the socioeconomic indicator but I think they could add a short notion of the very different distribution of the indicators as well. The occupational social class was defined as white-collar, manual, farmer, other self-employed and unknown. Is there a reason for not using such concepts as white-collar vs. blue-collar or non-manual vs. manual worker? Would it be possible to explain more about these two occupational social class categories? 2. The authors mention in the methods section that individuals were excluded from the study because they resided in institution (n=401). The number is not big but because dementia is one of the main drivers for institutionalization, I wish authors could clarify who were excluded individuals and if it may have impact on results. I assume that these individuals were only excluded from the analyses regarding the household income. 3. Reverse causality, suggesting that poor health leads to a lower socioeconomic position, may be related to cognitive decline but in midlife, I would consider mental issues, other morbidities and functional disabilities more likely reasons. Socioeconomic position was attained at the ages 53-57, which is why I think there are also other selection mechanisms than cognitive decline.
---

	4. The Figure 1 describes mortality differences nicely by age and socioeconomic indicator however, it is very difficult to compare age patterns, especially statistical significances between the panels a, b and c. Authors also elaborate on the differences between younger old and oldest old. It would help the reader to know approximately which age groups are discussed. Since age patterns are referred to a couple of times in the manuscript, my suggestion is to provide numbers in addition to the Figure in a (supplementary) table. 5. The authors suggest that it is a novel finding that they found socioeconomic inequality in dementia mortality in very old age (Page 12). From my point of view, the novelty of the study is related to showing inequality with three different socioeconomic indicators and the contribution of dementia mortality on inequalities in overall mortality. Please specify the novelty of the study more clearly. 6. It is not clear whether the last point in the section strength and limitations “we used individual death records..” is a strength or a limitation. Authors explain more about the death certificate in discussion but it is not clear for the reader why it would be problematic to identify dementia cases only from the death record. Please clarify. Minor comments:  1. In the abstract (and in the manuscript), please specify that the end of 2000 refers to the year 2000 2. Table 2 is two times, please correct the typo in the table 3 3. In Figure 1, it is difficult to separate two dotted lines, please make clearer if possible 4. In the caption for the Figure 1, was the follow-up for the mortality between 2001 and 2012 or until 2016 as stated in the abstract?
--	--

VERSION 1 – AUTHOR RESPONSE

Response to Reviewer #1:

This paper describes an analysis of Finnish register data, to determine socioeconomic inequalities – measured by education, occupation and income, in dementia incidence. The paper is well written and relevant. The analyses seem generally appropriate. Although the manuscript is already strong, I have listed a number of instances in which clarification or amendments may be needed.

#1. Page 4 line 36-39. Although the argument of the authors is correct (and is mentioned multiple times throughout the manuscript), I feel the authors should explicitly mention the crucial role of SEP in the selective survival effect. Those with a lower SEP tend to decrease earlier, and this is the reason that the surviving sample is more homogeneous in terms of education. Additionally, from the lower SEP groups, only the very healthiest survive to high ages, leading to a decrease in observed socioeconomic inequalities. This is often referred to as the “age-as-leveler” hypothesis.

Response: We thank the reviewer for pointing this out and we certainly agree that the role of socioeconomic position in the selective survival effect is important. We have now rephrased the text so that the role of socioeconomic position in selective survival is more explicit, and the text reads now:

“The lack of educational differentials among the oldest old may relate to selective survival. People with lower education experience higher mortality at younger ages, and those who survive to older

ages do so because of their better health. Thus, the population surviving to older ages is more homogeneous in terms of health-related characteristics and, as a result, the socioeconomic differences in mortality are diminished" (page 4, lines 36–46).

#2. I consider the disentangling of the contributions of education, occupation and income to dementia mortality as an important strength of this study. However it would benefit the introduction if the authors could elaborate on why this is important in the context of dementia; which different pathways are they thinking about with regards to dementia specifically? E.g., I can think of some specific links with education and with occupation (e.g., more complex work), but for income this is less clear. What effects would income have on dementia independent from education and occupation? The general argument on p.5, lines 3-8 is, in my opinion, not specific enough.

Response: We now specified the potential mechanisms linking income to dementia mortality in the introduction as follows: "A low household income may, in addition to material disadvantage, induce psychosocial stress, increasing the risk of dementia directly or through less favourable health behaviours. Disentangling the contributions of education, occupational social class and household income will thus provide important insights into the potential mechanisms how socioeconomic position shapes the risk of dementia death" (p. 5, lines 14–24). In order not to expand the length of the introduction too much, we focus on discussing the mechanisms in more detail in the discussion section of the paper (e.g. page 12, lines 31–34 on income; page 13, lines 7–23 on education; page 13, lines 33–40 on occupational social class).

#3. Why would the authors want to capture mediating pathways? It seems to be beyond the focus of this study to examine whether chronic health conditions and marital status would mediate the effect of SEP on dementia mortality. Moreover, no formal mediation analysis is performed. It is certainly important to adjust for confounding factors, and marital status might be a confounder because household rather than individual income is measured. But I don't see the added value of "adjusting for" chronic health conditions here. The authors might still regard them as potential confounders (perhaps because they suspect health selection effects), but then they must conceptualize them as such.

Response: We agree with the reviewer in that capturing mediating pathways was not the main focus of this paper, and that marital status and chronic health conditions may operate not only as mediators but also as confounders. In line with the reviewer's suggestion, we have now added more precision to the conceptualisation of these factors. We rephrased the wording in the introduction so that it now avoids strong causal language: "We further estimated models adjusting for other risk factors including marital status and various chronic health conditions" (page 5, lines 44–47).

By including chronic health conditions in the full model we observed that socioeconomic position was related to dementia mortality net of somatic health problems, and there were differences between the indicators of socioeconomic position. We found that the attenuation of excess dementia mortality was larger for household income than for education, which indicates that education has benefits for cognitive health above and beyond somatic health. To make the role of chronic health conditions more explicit in the text, we rephrased the text as follows:

"Impoverished material conditions may affect dementia risk through, for example, psychological stress[20] and health-related behaviours and cardiovascular risk factors.[16] Our findings show that the higher dementia mortality of the lowest household income quintiles was strongly – although not fully – related to greater morbidity and early retirement of these groups. It is also possible that severe health problems that were present already in midlife affected labour market participation and household incomes and thus confounded the association between income and the risk of dementia death. Future studies are needed to establish the causal relationship between these factors using formal mediation analysis techniques" (page 12, lines 31–48).

We also believe that because adjustment for traditional risk factors such as smoking, physical inactivity and cardiovascular disease is a strong convention in the literature, the inclusion of these conditions serves for better comparability between studies. Register data does not cover health behaviours, but we could take advantage of the rich health-care data to indicate these health-related factors. We added a notion of this data restriction in the Strengths and limitations of this study (page 3, lines 20–23) as well as in the discussion section (page 14, lines 28–35)

#4. Page 6 line 22-24. At first glance, the proportion of 20.7% of dementia deaths relative to all deaths seems very high. Could the authors elaborate more (either in the methods or discussion) on this number? Perhaps there are reasons for this (e.g., they counted primary, secondary and tertiary cause of death), but it is currently not clear how this incidence relates to incidences from other major studies.

Response: We thank the reviewer for pointing this out. The proportion of dementia deaths out of all deaths at the age of 70 and over is currently very high in Finland as in some other European countries. The proportion that we observed in our sample is in line with that reported in Finland at the population level (19% at the age of 65+ in 2014[1]) as well as in England and Wales (19% at the age of 80+ in 2016[2]). This is now mentioned in the discussion section as follows:

“Defined this way, we identified 21% of all deaths at the age of 70 and over to be attributable to dementia. This relatively high proportion is in line with that reported in England and Wales, where dementia accounted for 19% of all deaths at the age of 80 and over.[30]” (page 14, lines 17–22).

“Therefore, we believe our results are not biased by overreporting or underreporting of dementia as the cause of death[...]”(page 14, lines 26–28).

#5. Page 6, line 42. The authors state that there is also an education category “unknown”. But this category does not appear again in the manuscript, also not in the Tables. Where did it go?

Response: In Statistics Finland’s Register of Completed Education and Degrees, having no post-primary qualifications indicates that the individual has no more than basic education. There may be some situations where post-primary qualifications have not been registered, and concerns mainly immigrants. Because foreign-born individuals form a very small group in our sample (0.8%), any potential misclassification of education will not bring bias in our analyses. We specified the name of the education group as “basic education/no qualifications” (page 6, line 59), but because it in practice refers to having no more than basic education, we use the term “basic education” in the text thereafter.

#6. Page 6, line 46. How many individuals had no information on occupational social class at baseline?

Response: Information of occupational social class in the census year could only be determined for those who were employed at that time. The information was therefore lacking for 10 465 individuals. For 9942 individuals, the information could nevertheless be obtained from previous years in which the individuals were employed, leaving only 523 individuals with no occupation/unknown midlife occupational social class (this number reported in Table 1). We added this information in the methods section of the manuscript (page 7, lines 4–9), and named the category more precisely as “no occupation/unknown”.

#7. Page 6, line 51-53. The fact that no information on tax-free income transfers is included may be a major limitation, because it could lead to overestimation of socioeconomic inequalities. This should at least be discussed.

Response: We thank the reviewer for pointing this out, and we agree that information of disposable household income would better describe the actual resources in the household. However, for the census years 1970–1985 Statistics Finland only has information of household income subject to state taxation. Information of household disposable income was, however, available from year 1995 onwards, so we checked the correlation between taxable and disposable household incomes using information of 1995. The correlation was 0.97. We thus believe that the use of taxable income instead of disposable income has minor effects on the ranking of individuals in the income distribution, and because we used income quintiles, it very unlikely affects our main findings. However, like the reviewer suggested, this is something that should be discussed in the paper, and therefore we added the following text in the discussion section:

“Third, the information of household income was based on taxable income and the variable thus excludes certain monetary transfers such as housing allowance and social assistance. These means-tested sources of income may be especially relevant for people with health problems and those outside the labour market. This might lead to overestimation of the income effect. Information of disposable income was not available for years 1970–1985, but we carried out a robustness check for the correlation between taxable and disposable household incomes (as continuous variables) using the population aged 15 and over in 1995 and found the correlation to be as high as 0.97. Therefore, it is unlikely that the use of disposable income would change the ranking of individuals in the household income distribution to the extent that it would affect our main findings” (page 14, lines 38–57).

#8. Page 8. It may be relevant to not only work with rate differences (absolute differences in death rates) but also rate ratios (relative differences in death rates). This might provide important information, e.g. in terms of the development of inequalities across age groups. This could also better qualify the authors' statements about how the inequalities in dementia mortality compare to inequalities in overall mortality. I am not sure whether their present comparison takes into account that the dementia rates are generally much lower than overall mortality rates.

Response: We agree with the reviewer that presenting age-specific relative differences would strengthen the paper and the conclusions drawn. We revised table 2 so that it now includes also hazard ratios with their 95% confidence intervals. The contribution of dementia to overall socioeconomic mortality differentials was calculated using absolute rate differences, an approach employed also by previous studies.[3]

#9. Page 8. I am not sure if the method used by the authors to calculate the contribution of dementia to socioeconomic inequalities to overall mortality is the same or intended to do the same as calculating the Population Attributable Fraction (PAF)? If so, this could be mentioned in the manuscript.

Response: We analysed the contribution of dementia to overall socioeconomic differences in mortality, and not to overall mortality. Therefore, we applied a simple decomposition of mortality rates. It has a different interpretation than the population attributable fraction (i.e. the proportion of dementia deaths attributable to low socioeconomic position).

#10. Page 10 / Table 2. The figure that about 66% of socioeconomic inequalities in mortality above age 70 can be attributed to dementia deaths seems quite high to me (29.9% + 12.0% + 24.0%). Were the SEP indicators mutually adjusted here? Resonating my comment #4, are these surprising or unsurprising numbers in light of previous studies?

Response: Table 2 has now been formulated in a more precise way. The proportions do not add up to 100, because they are calculated for each socioeconomic indicator separately. We combined table 2 with the information of the (former) supplementary table in order to make it more readable and easier to interpret. In this set of analysis, the indicators of socioeconomic position are not mutually adjusted, because we addressed this question separately in the regression models presented in Table 3.

#11. Page 13, line 21. It is not clear what the authors mean with education “truly” enhancing brain health. Clarification and references could help here.

Response: We clarified the idea pursued in the sentence as follows: “Therefore, it is plausible that higher education enhances brain health and protects against (or postpones) not only the clinical symptoms but also the development of neurodegenerative disorders.” (page 13, lines 19–23).

#12. Page 13. Please discuss the high dementia death rate in the “unknown” occupation category.

Response: We added a paragraph with discussion on the occupational social class differences in dementia mortality, acknowledging the high hazard attached to having no occupation/unknown occupational social class. The text reads now:

“Occupational social class differences in dementia mortality were modest following adjustment for education and household income. In particular, the high hazard among those with no occupation disappeared after these adjustments indicating that this group experienced multiple socioeconomic disadvantages” (page 13, lines 26–33).

#13. Discussion. It would be valuable if the implications of finding that different socioeconomic indicators are independently related to dementia mortality are discussed. What different pathways could they indicate? What does this mean for prevention or intervention?

Response: We have now more clearly discussed the potential mechanisms related to each socioeconomic indicator. We elaborated the discussion on pathways as follows:

“Impoverished material conditions may affect dementia risk through, for example, psychological stress[20] and health-related behaviours and cardiovascular risk factors.[16] Our findings show that the higher dementia mortality of the lowest household income quintiles was strongly – although not fully – related to greater morbidity and early retirement of these groups” (page 12 lines 31–38).

“The results suggest, nevertheless, that higher social class occupations may involve greater cognitive demands and intellectual engagement, and thus enhance cognitive health.[26,27] In contrast, lower class occupations or long periods of economic inactivity due to unemployment or early retirement may reduce opportunities for cognitive investment” (page 13 lines 33–40).

We also expanded on the implications of the results as follows: “The results indicate that dementia mortality may be amenable to socioeconomic interventions in midlife” (page 15 lines 39–41).

#14. Table 1. For clarification, please add to the Table title that SEP was measured at about age 55.

Response: We thank the reviewer for this useful suggestion. We now specified in all table legends and the figure caption that the indicators of socioeconomic position refer to midlife socioeconomic position and were measured from the population censuses of 1970–1985, at the age of 53–57.

#15. Table 3. Table title says Table 2, should be 3.

Response: We thank the reviewer for pointing this out. We have now corrected the numbering of Table 3.

Response to Reviewer #2:

Thank you for inviting me to review the manuscript “Midlife socioeconomic position and old-age dementia mortality: a large prospective register-based study from Finland”. The study aimed to examine the association of three socioeconomic position indicators with dementia-related mortality

and the contribution of dementia to overall mortality differences at older ages. The study showed that in the Finnish older population, all three socioeconomic position indicators were independently associated with dementia mortality (higher dementia mortality in lower socioeconomic positions), dementia accounted 25-30% of the total mortality between high and low educated and according to household income but less according to occupational social class in 70+ population, and that the contribution of dementia on socioeconomic mortality differences increased with age.

This study adds to the current knowledge that dementia is an important contributor of socioeconomic inequality in old age mortality. Regarding the rapid increase in the oldest old population in Finland and globally, this study is timely and provides new information as well as strengthen already reported socioeconomic inequality in dementia mortality in the oldest old in Finland. The study is well conducted however, I have few comments and questions for clarifications for the authors.

General comments:

1. The study uses three socioeconomic position indicators namely education, occupational social class and household income, which all have more than two categories. Could authors provide an explanation why analyses were conducted contrasting only, so to say, extreme ends? Authors elaborate in the discussion section on the found differences in results based on the socioeconomic indicator but I think they could add a short notion of the very different distribution of the indicators as well. The occupational social class was defined as white-collar, manual, farmer, other self-employed and unknown. Is there a reason for not using such concepts as white-collar vs. blue-collar or non-manual vs. manual worker? Would it be possible to explain more about these two occupational social class categories?

Response: The choice to contrast the high and low education groups, non-manual and manual employees and the highest and lowest income quintiles was a practical one; plotting the Kaplan-Meier survival functions for all categories would make the figures rather messy and difficult to read. However, we conducted the Cox regression analyses using the information of all categories in each indicator (table 3), therefore showing the associations not only for the extremes but for the other groups as well.

We also acknowledge the different distributions of the indicators, and show these in Table 1. We now more clearly refer to the different distributions in the introduction (page 4 lines 48–53), and we added a notion in the discussion section on the implications of the skew education distribution: “Also, given the small proportion of people with tertiary education in these cohorts (10%), it is possible that this forms a select group with multiple advantages including higher childhood socioeconomic position and early cognitive ability” (page 15 lines 11–16).

We thank the reviewer for pointing out the conceptualisation of occupational social class, and we have now corrected the category names as non-manual and manual.

2. The authors mention in the methods section that individuals were excluded from the study because they resided in institution (n=401). The number is not big but because dementia is one of the main drivers for institutionalization, I wish authors could clarify who were excluded individuals and if it may have impact on results. I assume that these individuals were only excluded from the analyses regarding the household income.

Response: 401 individuals did not belong in the household population in the census year, and for that reason they did not have information on household income. This is most commonly due to institutional residence. To have the risk population consistent across all analyses, we excluded these individuals from the sample. However, institutionalization due to dementia at around the age of 55 is very uncommon, so we believe this does not affect our findings. Individuals who were institutionalized after the census year (before, on or after the baseline year of 2000) were included in all analyses, and thus institutionalization due to dementia does not affect the results. We rephrased the text as follows: "Individuals with missing census information due to residing outside of Finland (n=920) and those with missing household income information due to not belonging in the household population in the census year (n=401) were excluded" (page 6 lines 20–22).

3. Reverse causality, suggesting that poor health leads to a lower socioeconomic position, may be related to cognitive decline but in midlife, I would consider mental issues, other morbidities and functional disabilities more likely reasons. Socioeconomic position was attained at the ages 53-57, which is why I think there are also other selection mechanisms than cognitive decline.

Response: We thank the reviewer for pointing this out and we certainly agree that a lower socioeconomic position in midlife may be related to morbidities other than dementia and functional disabilities. This is now mentioned in the text as follows: "It is also possible that severe health problems that were present already in midlife affected labour market participation and household incomes and thus confounded the association between income and the risk of dementia death" (page 12 lines 38–43).

4. The Figure 1 describes mortality differences nicely by age and socioeconomic indicator however, it is very difficult to compare age patterns, especially statistical significances between the panels a, b and c. Authors also elaborate on the differences between younger old and oldest old. It would help the reader to know approximately which age groups are discussed. Since age patterns are referred to a couple of times in the manuscript, my suggestion is to provide numbers in addition to the Figure in a (supplementary) table.

Response: We have now added Supplementary Table 2 with the Kaplan-Meier survival function estimates at specific ages (70, 75, 80, 85, 90, 95 and 100 years) and their 95% confidence intervals. Also, the relative differences in Table 2 indicate now more clearly the differences by specific age groups (70–79, 80–89 and 90 and over). In addition to these, we added in the discussion section the age groups when referring to younger old and older old (page 11, line 56; page 12 lines 24 and 27).

5. The authors suggest that it is a novel finding that they found socioeconomic inequality in dementia mortality in very old age (Page 12). From my point of view, the novelty of the study is related to showing inequality with three different socioeconomic indicators and the contribution of dementia mortality on inequalities in overall mortality. Please specify the novelty of the study more clearly.

Response: We agree that the novelty of the study is also related to showing inequality with three different socioeconomic indicators and to the contribution of dementia mortality to inequalities in overall mortality. This is now specified in the discussion as follows (new parts underlined):

"In this study we have shown that dementia mortality at older ages is socioeconomically patterned in terms of multiple indicators of socioeconomic position" (page 11 lines 25–27).

“Overall, the results of this study suggest that all three indicators of socioeconomic position are important factors in bringing about socioeconomic differences in dementia mortality, also influencing inequalities in overall mortality among the older population” (page 13 lines 40–45).

“This study provides new insight into the socioeconomic inequalities in old-age mortality by showing a consistent pattern in dementia mortality by multiple indicators of socioeconomic position” (page 15 lines 25).

We also specified that the novel finding of differences at the oldest ages is related to education in particular (page 12 line 7).

6. It is not clear whether the last point in the section strength and limitations “we used individual death records..” is a strength or a limitation. Authors explain more about the death certificate in discussion but it is not clear for the reader why it would be problematic to identify dementia cases only from the death record. Please clarify.

Response: We have now clarified the point on identifying dementia from the death register as follows: “Dementia is documented in the national death register with high specificity” (page 3 line 18).

Minor comments:

1. In the abstract (and in the manuscript), please specify that the end of 2000 refers to the year 2000

Response: We have now corrected the wording (page 2 line 19; page 6, line 13).

2. Table 2 is two times, please correct the typo in the table 3

Response: We have corrected the numbering of Table 3 (see also response to Reviewer 1, comment #15).

3. In Figure 1, it is difficult to separate two dotted lines, please make clearer if possible

Response: We replaced the dotted lines with coloured solid lines to make it easier for the reader to separate between the different groups.

4. In the caption for the Figure 1, was the follow-up for the mortality between 2001 and 2012 or until 2016 as stated in the abstract?

Response: The follow-up time was from 2001 to 2016, and we corrected this in the Figure 1 caption. We thank the reviewer for spotting this error.

VERSION 2 – REVIEW

REVIEWER	Almar Kok Amsterdam UMC, location VUmc, the Netherlands
REVIEW RETURNED	10-Oct-2019

GENERAL COMMENTS	I compliment the authors for their clear and adequate response to my comments. Most issues have been sufficiently addressed, except for a few points that, in my view, require further consideration. #1. In my previous comment #3 I pointed out that capturing mediating pathways is beyond the scope of this paper and that these 'mediators' could also be acting as confounders of the SEP - > dementia relationship. For these reasons, I did not see the added value of these variables for the present paper. I still think that the added value and the role of these variables are unclear. Specifically, the authors responded that they "added more precision" to the conceptualisation of these factors and made the role of chronic health conditions "more explicit in the text". In my opinion, the opposite is true. In the original manuscript, the authors were at least explicit about conceptualizing marital status and chronic conditions as mediators. But in the revised manuscript, they now avoid causal language, which actually obscures their reasons for adding marital status and chronic conditions to the analyses. For example, the new sentences "We further estimated models adjusting for..." and "education has benefits for cognitive health above and beyond somatic health" do not clarify the (assumed) role of the risk factors in their models; are they mediators or confounders? The phrase "was strongly – although not fully – related to greater morbidity..." is equally ambiguous. What does "related to" mean? Furthermore, the authors argue that adding "traditional risk factors" such as chronic diseases is a strong convention in the literature. I disagree. These factors are typically introduced in an analysis of socioeconomic inequalities in health (SEIH) only if a) researchers have strong suspicions of health selection effects and want to control for such effects; or b) researchers aim to examine pathways or mechanisms of SEIH (mediation effects). In my view, aim b) is beyond the scope of the paper, with which the authors agree. At the same time, the case for mediation is perhaps stronger than for health selection, as the risk factors have been measured years after the measure of SEP. And a) also does not seem to be in line with the authors' expectations. In sum, the role of the risk factors in this paper is still ambiguous. I am not sure what the best solution to this is. On the one hand, if the authors still wish to retain the risk factors in their analysis (which in my view is not crucial), it would be most consistent to conceptualize them as potential confounders, because examining mediation is beyond the aim of this study, and one might ask why only these two potential mediating factors were examined and not others. On the other hand, the case for conceptualizing the risk
--

	factors as confounders (health selection) is not very strong, as these factors have been measured years after the measurement of SEP. In any case, if the authors wish to retain the risk factors in their analysis, this should be accompanied by a clearer viewpoint as to their assumed causal role in the analysis, stronger arguments for their inclusion and not other factors that might be mediators, and a clearer reflection on the limitations. #2. I found the authors' response to #5 unclear. Are they saying that the "unknown" category consists of participants for whom their educational qualifications have not been registered, and these are now categorized as "basic education"? The authors appear to claim that "in practice", 'unknown' is the same as having no more than basic education, but on what empirical data is this claim based? If the education is unknown and it cannot be derived from other data, it should remain 'unknown' or missing in the analysis. #3. In response to comment #7, the authors report a correlation between taxable and disposable income of 0.97 in the population aged 15 and over in 1995. To what extent can this be generalized to the population of 70 and over included in their analysis? It might not be the same, as in old age, taxable income might be much lower compared to disposable income.
--	--

REVIEWER	Linda Enroth Faculty of Social Sciences (Health Sciences) and Gerontology Research Center, Tampere University, Finland
REVIEW RETURNED	15-Oct-2019

GENERAL COMMENTS	The authors made great effort to improve the manuscript. Broadening the introduction with mechanisms linking SEP to dementia and editing the results (e.g. providing numbers in addition to figure and Cox-models) clarified the main aim of the study.
---

VERSION 2 – AUTHOR RESPONSE

Response to Reviewer #1:

I compliment the authors for their clear and adequate response to my comments. Most issues have been sufficiently addressed, except for a few points that, in my view, require further consideration.

Response: We thank the reviewer for the positive feedback.

#1. In my previous comment #3 I pointed out that capturing mediating pathways is beyond the scope of this paper and that these 'mediators' could also be acting as confounders of the SEP -> dementia relationship. For these reasons, I did not see the added value of these variables for the present paper. I still think that the added value and the role of these variables are unclear.

Specifically, the authors responded that they “added more precision” to the conceptualisation of these factors and made the role of chronic health conditions “more explicit in the text”. In my opinion, the opposite is true. In the original manuscript, the authors were at least explicit about conceptualizing marital status and chronic conditions as mediators. But in the revised manuscript, they now avoid causal language, which actually obscures their reasons for adding marital status and chronic conditions to the analyses.

For example, the new sentences “We further estimated models adjusting for...” and “education has benefits for cognitive health above and beyond somatic health” do not clarify the (assumed) role of the risk factors in their models; are they mediators or confounders? The phrase “was strongly – although not fully – related to greater morbidity...” is equally ambiguous. What does “related to” mean?

Furthermore, the authors argue that adding “traditional risk factors” such as chronic diseases is a strong convention in the literature. I disagree. These factors are typically introduced in an analysis of socioeconomic inequalities in health (SEIH) only if a) researchers have strong suspicions of health selection effects and want to control for such effects; or b) researchers aim to examine pathways or mechanisms of SEIH (mediation effects).

In my view, aim b) is beyond the scope of the paper, with which the authors agree. At the same time, the case for mediation is perhaps stronger than for health selection, as the risk factors have been measured years after the measure of SEP. And a) also does not seem to be in line with the authors’ expectations.

In sum, the role of the risk factors in this paper is still ambiguous. I am not sure what the best solution to this is. On the one hand, if the authors still wish to retain the risk factors in their analysis (which in my view is not crucial), it would be most consistent to conceptualize them as potential confounders, because examining mediation is beyond the aim of this study, and one might ask why only these two potential mediating factors were examined and not others. On the other hand, the case for conceptualizing the risk factors as confounders (health selection) is not very strong, as these factors have been measured years after the measurement of SEP.

In any case, if the authors wish to retain the risk factors in their analysis, this should be accompanied by a clearer viewpoint as to their assumed causal role in the analysis, stronger arguments for their inclusion and not other factors that might be mediators, and a clearer reflection on the limitations.

Response: We thank the reviewer for the thorough and insightful comment. Based on this, we have reconsidered the conceptualization of the included risk factors. We believe that our measures of chronic health conditions reflect confounding factors in that they have most likely developed over a long period of time and thus reflect health behaviours or health problems already present in midlife. For example, developing a chronic alcohol-attributable disease by old age will have taken many years of harmful alcohol consumption to the extent that it may have affected midlife incomes. However, we believe that stroke is an exception to this, as stroke has more direct and immediate consequences for dementia risk. In order to provide a clear conceptualisation of chronic health conditions as confounders, we have now removed stroke from the list of covariates and have estimated the full

model adjusting for the remaining chronic health conditions, i.e. alcohol-related diseases and alcohol poisoning, asthma and other chronic obstructive pulmonary diseases, diabetes, and heart disease.

We have now conceptualised the chronic health conditions as confounders in the text. We also present the justification in the Methods section: “These chronic conditions may confound the association between midlife SEP and dementia mortality as the diseases usually develop over a long period of time and thus reflect health behaviours or health problems already present in midlife.” (page 7, lines 52–59).

We cannot, however, exclude the possibility that some of the chronic health conditions may have developed as a response to socioeconomic disadvantage, especially since the conditions were measured after the measurement of midlife socioeconomic position. In this case, our estimates in the full model would be conservative as they would over-adjust part of the effect of socioeconomic disadvantage. We reflect on these limitations and acknowledge the possibility of mediation in the discussion section as follows:

“Individuals in the lowest income quintile represent the most disadvantaged population subgroups with multiple potential dementia risk factors. Our findings show that the higher dementia mortality of the lowest household income quintiles was strongly — although not fully — confounded by greater morbidity of these groups. Severe health problems that were already present in midlife have potentially affected both household incomes and the risk of dementia death. However, we cannot rule out the possibility of mediation, especially because chronic health conditions were measured after midlife income; impoverished material conditions may also affect dementia risk through, for example, health-related behaviours, cardiovascular risk factors [16], and psychological stress.[20] In the presence of mediation, our estimates would be conservative as they would overadjust part of the effect of socioeconomic disadvantage. Future studies are needed to establish the causal relationship between mediating factors and dementia mortality using mediation analysis techniques” (page 12, lines 34–57).

#2. I found the authors' response to #5 unclear. Are they saying that the “unknown” category consists of participants for whom their educational qualifications have not been registered, and these are now categorized as “basic education”? The authors appear to claim that “in practice”, 'unknown' is the same as having no more than basic education, but on what empirical data is this claim based? If the education is unknown and it cannot be derived from other data, it should remain 'unknown' or missing in the analysis.

Response: We may have stated the justification of this categorization unclearly partly because the same categorisation of education using Finnish register data has been extensively used also in previous studies (e.g.,[2] and [3]).

The oldest birth cohort included in our analyses were born in 1913 and were 8 years old in 1921 when the Compulsory School Attendance Act came into force in Finland. Based on the law, all children aged 7–13 years were to receive primary education. By mid-1930s, about 90% of the 7–15-year-old population received schooling.[1] Thus, it is reasonable to expect that all individuals in our analyses have received at least basic education.

If an individual has no entry in the Register of Completed Education and Degrees, it means that they have no post-primary qualifications. This is because the education register only records qualifications from secondary-level education or higher (ISCED 1997 3–6). As a result, the ‘basic education/no qualifications’ category consists of those who have no registered post-primary qualifications, but who can be expected to have basic education.

The potential misclassification that we mentioned in the previous response related to immigrants for whom post-primary education might not be registered as completely (as it is for the Finnish-born

population) even though they might have received post-primary qualifications in their country of origin. And vice versa, we cannot know if they have received basic education in their country of origin. However, the proportion of immigrants in our sample is very small (0.8%), so this will not affect our results (in fact, a larger proportion of the non-Finnish-born had post-primary qualifications compared to the Finnish-born in our sample).

#3. In response to comment #7, the authors report a correlation between taxable and disposable income of 0.97 in the population aged 15 and over in 1995. To what extent can this be generalized to the population of 70 and over included in their analysis? It might not be the same, as in old age, taxable income might be much lower compared to disposable income.

Response: We thank the reviewer for this remark. We have now calculated the correlation between taxable and disposable household income in the population aged 53–57 (the same age group as for whom income was measured in our study) in 1995. The correlation is 0.98, and we added this information in the discussion section of the manuscript (page 14, line 59).

Response to Reviewer #2:

The authors made great effort to improve the manuscript. Broadening the introduction with mechanisms linking SEP to dementia and editing the results (e.g. providing numbers in addition to figure and Cox-models) clarified the main aim of the study.

Response: We thank the reviewer for the positive comments and all the work done to improve the quality of the manuscript.

References (used in the response letter)

- 1 Statistics Finland - Education in Finland: more education for more people. https://www.stat.fi/tup/suomi90/marraskuu_en.html (accessed 4 Nov 2019).
- 2 Hoffmann R, Kröger H, Tarkiainen L, et al. Dimensions of Social Stratification and Their Relation to Mortality: A Comparison Across Gender and Life Course Periods in Finland. *Social Indicators Research* 2019;:1–17.
- 3 Kilpi F, Martikainen P, Konttinen H, et al. The spillover influence of partner's education on myocardial infarction incidence and survival. *Epidemiology* 2018;29:237–245.

VERSION 3 - REVIEW

REVIEWER	Almar Kok Amsterdam UMC, the Netherlands
REVIEW RETURNED	02-Dec-2019

GENERAL COMMENTS	The authors have adequately addressed my remaining concerns. I have no further comments.
--